# Bcl-xL Is Spontaneously Inserted into Preassembled Nanodiscs and Stimulates Bax Insertion in a Cell-Free Protein Synthesis System

**DOI:** 10.3390/biom13060876

**Published:** 2023-05-23

**Authors:** Akandé Rouchidane Eyitayo, Axel Boudier-Lemosquet, Stéphane Chaignepain, Muriel Priault, Stéphen Manon

**Affiliations:** 1Institut de Biochimie et de Génétique Cellulaires, Université de Bordeaux, CNRS, UMR 5095, 33077 Bordeaux, France; akande.rouchidane-eyitayo@ibgc.cnrs.fr (A.R.E.); axel.boudier@ibgc.cnrs.fr (A.B.-L.); stephane.chaignepain@u-bordeaux.fr (S.C.); 2Centre de Génomique Fonctionnelle de Bordeaux, Université de Bordeaux, 33077 Bordeaux, France

**Keywords:** Bcl-xL, Bax, nanodiscs, cell-free synthesis, membrane insertion, MALDI, apoptosis

## Abstract

The antiapoptotic protein Bcl-xL is a major regulator of cell death and survival, but many aspects of its functions remain elusive. It is mostly localized in the mitochondrial outer membrane (MOM) owing to its C-terminal hydrophobic α-helix. In order to gain further information about its membrane organization, we set up a model system combining cell-free protein synthesis and nanodisc insertion. We found that, contrary to its proapoptotic partner Bax, neosynthesized Bcl-xL was spontaneously inserted into nanodiscs. The deletion of the C-terminal α-helix of Bcl-xL prevented nanodisc insertion. We also found that nanodisc insertion protected Bcl-xL against the proteolysis of the 13 C-terminal residues that occurs during expression of Bcl-xL as a soluble protein in *E. coli*. Interestingly, we observed that Bcl-xL increased the insertion of Bax into nanodiscs, in a similar way to that which occurs in mitochondria. Cell-free synthesis in the presence of nanodiscs is, thus, a suitable model system to study the molecular aspects of the interaction between Bcl-xL and Bax during their membrane insertion.

## 1. Introduction

Apoptotic programmed cell death is controlled by proteins of the Bcl-2 family (Bcl-2s). These proteins are functionally divided into three subfamilies: antiapoptotic proteins (e.g., Bcl-2 and Bcl-xL), multidomain proapoptotic proteins (Bax, Bak, and Bok), and BH3-only proteins (e.g., Bid, Bim, and Bad) acting as modulators of the two other subfamilies. Altogether, the main function of Bcl-2s is to regulate the permeability of the mitochondrial outer membrane (MOM) to mitochondrial apoptogenic factors (e.g., cytochrome c and Smac/Diablo) that are released during the early steps of apoptosis. Typically, multidomain proapoptotic proteins Bax and Bak form large pores in the MOM, after being activated by BH3-only proteins such as tBid and Bim. This activation is counteracted by antiapoptotic proteins Bcl-2 and Bcl-xL through their physical interactions with Bax and Bak. This physical interaction can be prevented by other BH3-only proteins such as Bad (see [1,2,3,4] for recent reviews on the function and regulation of Bcl-2s).

A remarkable feature of Bcl-2s is the structural homology that exists between pro- and antiapoptotic proteins. Beyond the sequence homology that characterizes this family in the “Bcl-2 homology domains (BH)”, structural data have shown that the soluble conformations of anti- and proapoptotic proteins are very similar (see [5,6,7,8] for reviews). However, this similarity cannot be extended to their membrane conformation. Indeed, proapoptotic proteins Bax and Bak are known to form large oligomers [9,10,11,12] that define a pore large enough to allow the passage of apoptogenic factors having sizes between ~3.2 and 4.5 nm [13,14]. This involves several conformational changes that radically reshape the proteins [15,16,17,18]. On the contrary, Bcl-2 and Bcl-xL are expected to be anchored in the membrane under the form of monomers through their C-terminal hydrophobic α-helix, with moderate, albeit significant, changes in the conformation of the protein core [19,20,21,22].

Both pro- and antiapoptotic members of this family are notably difficult to purify, due to the presence of their C-terminal hydrophobic α-helix and to the moderate capacity of some detergents to stabilize oligomers. Most structural and functional data obtained on purified protein before the year 2000 were obtained on proteins deprived of the C-terminal hydrophobic α-helix. In 2000, recombinant Bax was obtained as an auto-cleavable fusion with intein, which allowed the purification and the NMR structure resolution of the full-length and untagged soluble protein [23]. For long, recombinant Bcl-xL was produced as a soluble protein in *E. coli* and was considered as being “full-length”. However, in 2015, Yao et al. demonstrated that the soluble fraction of Bcl-xL was cleaved between residues 218 and 219, thus removing a large part of the C-terminal α-helix, while the full-length protein was present in inclusion bodies [21]. Structural NMR data on membrane-anchored vs. soluble Bcl-xL led to a reconsideration of the role of the C-terminal α-helix of Bcl-xL, questioning its function as a “passive” membrane anchor. Indeed, long-range conformational changes could connect the membrane insertion of the C-terminal α-helix to movements of other domains of the protein, such as the BH3 domain or the α1–α2 loop [22,24]. In addition to its function in Bcl-xL insertion, the C-terminal α-helix of Bcl-xL also has a function in the (co)-retrotranslocation of Bax by Bcl-xL from the MOM to the cytosol [25,26]. Furthermore, post-translational modifications such as phosphorylation [27,28] and deamidation [29,30] regulate the pro-survival functions of Bcl-xL.

Investigations on how Bcl-xL is addressed and inserted into membranes require setting up a robust model that minimizes the time between protein synthesis and insertion. Cell-free protein synthesis in the presence of membrane models such as liposomes or nanodiscs (ND) meets this requirement [31,32,33]. In the present study, we show that full-length Bcl-xL can be simultaneously synthesized and inserted into nanodiscs, which further stimulates Bax insertion, thus representing a suitable model for further biochemical and structural studies.

## 2. Materials and Methods

### 2.1. Cloning of Full-Length Bcl-xL and Bax

Full-length Bcl-xL and Bax were cloned between the Nde1 and Xho1 sites of the piVex 2.3MCS plasmid. A stop codon was introduced to exclude the His_6_-tag. The Bcl-xLΔC(213) mutant was constructed by the Quickchange method with the Phusion polymerase (Thermofisher, Illkirch, France).

### 2.2. Production of the Scaffold Protein His7-MSP1E3D1

The scaffold protein His_7_-MSP1E3D1 was produced as described previously [34]. The plasmid pET28b encoding His_7_-MSP1E3D1 (Addgene, Watertown, MA, USA) was transformed into *E. coli* BL21DE3 and plated on LB solid medium (0.5% yeast extract, 1% tryptone, 1% NaCl, and 1.5% agar) containing 30 mg/L kanamycin. One isolated colony was pre-grown in LB medium containing 30 mg/L kanamycin overnight at 37 °C. The preculture was inoculated into 1 L of TB medium (2.4% yeast extract, 1.2% tryptone, 0.5% glycerol, and 8.9 mM sodium phosphate buffer; pH 7.0) containing 30 mg/L of kanamycin at OD_600_ = 0.2 and allowed to grow until OD_600_ = 0.9. His_7_-MSP1E3D1 expression was induced by adding 1 mM IPTG, and the growth was extended at 37 °C until OD_600_ reached 1.8. Cells were pelleted by centrifugation (4000× *g*, 10 min, 4 °C, Sorvall SLA1500 rotor), and then resuspended in 30 mL of lysis buffer (100 mM Na_2_HPO_4_, pH 7.6, 50 mM NaCl, 1 mM PMSF, 1 mM Pic8340 (Proteases inhibitors, Sigma, Saint-Quentin Fallavier, France), 1 mM AEBSF, 1% Triton X100, and one tablet of Complete^TM^ EDTA-free). Cells were lysed by sonication (50% pulsation, Power on 7 (max microTip), seven cycles of 30 s of sonication followed by 30 s pause). Cell debris was pelleted by centrifugation (46,000× *g*, 30 min, 4 °C, Beckman 45Ti rotor). The supernatant was loaded overnight (in a closed circuit on a peristaltic pump) on a Ni-NTA column (HisTrap FF 5 mL, Cytiva, Vélizy-Villacoublay, France) previously equilibrated in lysis buffer. The circuit was opened, and the column was washed sequentially with five volumes of Buffers 1–3 (Buffer 1: 40 mM Tris/HCl pH 8.0, 300 mM NaCl, and 1% Triton X100; Buffer 2: 40 mM Tris/HCl pH 8.0, 300 mM NaCl, and 50 mM sodium cholate; Buffer 3: 40 mM Tris/HCl pH 8.0, 300 mM NaCl, and 50 mM imidazole). The column was then connected to an Äkta purifier and washed with one more volume of Buffer 3. Proteins were eluted with elution buffer (40 mM Tris/HCl pH 8.0, 300 mM NaCl, and 300 mM imidazole). The eluate was then dialyzed against the desalting buffer (10 mM Tris/HCl pH 8.0, and 100 mM NaCl) and concentrated with a Vivaspin concentrator (10 kDa cutoff, Sartorius, Göttingen, Germany). The protein (~7 mg/mL) was stored as working aliquots at −80 °C.

### 2.3. Nanodisc Formation

Nanodisc preparation was performed as previously described [34]. Briefly, 2.5 mg of 1-palmitoyl-2-oleoyl-sn-glycero-3-phosphocholine (POPC) in chloroform (Anatrace, Maumee, OH, USA) was dried for 3 h under argon flow. The dry lipid film was resolubilized in nanodisc buffer (10 mM Tris/HCl, pH 8.0, 100 mM NaCl, and 200 mM sodium cholate). Then, 1.5 mg of His_7_-MSP1E3D1 was added and left for equilibration for 1 h. Detergent was removed by adding 150 mg of BioBeads SM-2 (Bio-Rad, Marnes-la-Coquette, France) for 3 h. The sample was then centrifuged at 20,000× *g* for 15 min at 4 °C, filtered through a 0.22 µm membrane, and loaded on a SEC column (Superdex 200 10/300GL, Cytiva). SEC was run in the same buffer (without cholate) at 0.5 mL per minute. When needed, nanodiscs were concentrated on Vivaspin with a 30 kDa cutoff (final concentration: 0.2 to 0.4 mM His_7_-MSP1E3D1).

Several experiments were conducted with phospholipid mixtures such as *E. coli* polar extract (Avanti 100600C) that mostly contains phosphatidylethanolamine (PE), phosphatidylglycerol (PG) cardiolipin (CL), and a “mitochondria-like” mixture of POPC, POPE, POPG, POPS, PI, and CL, as described previously [35].

### 2.4. Cell-Free Protein Synthesis and Protein Insertion into Nanodiscs

Cell-free protein synthesis was conducted as described previously [34] in a small-volume dialysis chamber (100 µL), separated from a feeding reservoir (1700 µL) by a dialysis membrane (MW 10,000 Da). The system was routinely set up in inverted microcentrifuge tubes with the dialysis membrane stuck between the cap containing the reaction mix and the bottom-cut tube containing the feeding mix. Both chambers contained 0.1 M K/HEPES (pH 8.0), 1 mM EDTA, 0.05% NaN_3_, 2% PEG 8000, 151 mM potassium acetate, 7.1 mM magnesium acetate, 0.1 mg/mL folinic acid (calcium salt), 2 mM DTT, 1 mM (each) NTP mix, 0.5 mM (each) aminoacid mix, 1 mM (each) RDEWCM mix, 20 mM K/PEP, 20 mM acetylphosphate (Li/K salt), and protease inhibitor cocktail (Complete without EDTA, Roche, Basel, Switzerland). The reaction mix was supplemented with components for the synthesis: 35% (*v*/*v*) S30 *E. coli* BL21DE3 lysate (in 10 mM Tris/acetate buffer, pH 8.2, 14 mM magnesium acetate, 0.6 mM potassium acetate, and 0.5 mM DTT), 0.04 mg/mL pyruvate kinase (Sigma), 15 µg/mL pIVEX plasmid, 0.5 mg/mL tRNA mix (Roche), 6 units/mL T7 RNA polymerase [36] (for the preparation of bacterial lysate and T7 RNA polymerase), and 3 units/mL RNasin (Promega, Charbonnières-les-Bains, France). Depending on the experiments (see results), preformed nanodiscs were added. The synthesis was conducted for 16 h (overnight) at 28 °C under gentle agitation. The reaction mix was centrifuged for 15 min at 20,000× *g* to separate the pellet containing precipitated proteins and the supernatant containing nanodiscs and soluble proteins.

Nanodiscs were purified from supernatants by adding 100 µL of Ni-NTA Sepharose (Qiagen, Courtaboeuf, France) prewashed in the nanodisc buffer (10 mM Tris/HCl, pH 8.0, and 100 mM NaCl) and incubated at 4 °C for 15 min. Ni-NTA Sepharose beads were then washed thrice by centrifugation/resuspension with 500 µL of Tris/HCl 10 mM (pH 8.0) and NaCl 500 mM buffer; bound nanodiscs were then eluted with Tris/HCl 10 mM (pH 8.0), NaCl 100 mM, and imidazole 300 mM.

### 2.5. MALDI-MS Analysis

For MALDI-MS analyses, Bcl-xL-containing nanodiscs were added as above to Ni-NTA Sepharose (Qiagen) and washed thrice with 10 mM Tris/HCl (pH 8.0) and 500 mM NaCl. Beads were then placed in IP columns (Pierce, Appleton, WI, USA) before adding 200 µL of the same buffer supplemented with 50 mM sodium cholate to solubilize nanodiscs. Eluted Bcl-xL was then dialyzed thrice against Tris/HCl 20 mM (pH 7.2), NaCl 20 mM, and DTT 5 mM (preventing dimer formation) to eliminate cholate, and then concentrated on Vivaspin (10 kDa cutoff, Sartorius).

Recombinant Bcl-xL expressed in *E. coli* was purified from inclusion bodies, according to [21,37]. The protein was stored in Tris/HCl 20 mM (pH 8.0), NaCl 450 mM, and DTT 10 mM. Before MALDI-MS analyses, the protein was dialyzed twice against Tris/HCl 20 mM (pH 7.2), NaCl 20 mM, and DTT 5 mM.

Dialysis steps are critical to reduce the amount of cholate and NaCl that could later interfere with the co-crystallization of the sample with the matrix. Samples were mixed (1:1 *v*/*v*) with a matrix solution of sinapic acid (10 mg/mL) in 1:1 (*v*/*v*) acetonitrile/deionized water with 0.1% trifluoroacetic acid. Each mixture (2 µL thereof) was dropped on the MALDI target plate using the dried-droplet method. MALDI-MS analyses were carried out on a MALDI-TOF mass spectrometer (Ultraflex III, Bruker, Billerica, MA, USA) used in linear mode. Calibration was achieved using singly charged cytochrome c and myoglobin ions ( [M + H] + average = 12,361.0 Da and [M + H] + average = 16,952.3 Da, respectively) and the gas phase dimer of these proteins ([2M + H] + average = 24,720.9 Da and [2M + H] + average = 33,903.6 Da, respectively). The mass spectra were acquired by averaging 8000 shots. The spectra were processed (including background subtraction and smoothing) using FlexAnalysis (Bruker). Experimental masses are the averages of two independent measurements.

### 2.6. Miscellaneous

Tris/glycine and taurine/imidazole SDS-PAGE and Western blots were performed as described previously [38]. Bcl-xL was revealed with rabbit monoclonal antibody E18 (Abcam), His_7_-tag was revealed with mouse monoclonal antibody 4E3D10H2/E3 (Thermofisher), and Bax was revealed with mouse monoclonal antibody 2D2 (Santa-Cruz, Santa Cruz, CA, USA). HRP-coupled secondary antibodies were from Jackson Laboratories, and blots were revealed with Immobilon Forte (Millipore, Burlington, MA, USA). Images were captured with a G-Box digital camera (Syngene, Cambridge, UK) and processed with Image J 1.48v (https://imagej.nih.gov/ij/index.html, downloaded 24 May 2019).

## 3. Results

### 3.1. Neosynthesized Bcl-xL Is Spontaneously Inserted into Nanodiscs

It is widely acknowledged that Bcl-xL, whether at endogenous levels or overexpressed, is essentially localized in the mitochondrial outer membrane [39]. This is likely due to the presence of a stretch of positively charged residues (termed the “X domain”) located immediately upstream the C-terminal hydrophobic helix, that may stabilize its insertion by interacting with negatively charged lipid polar heads. Indeed, suppressing these positive charges decreased the mitochondrial localization of Bcl-xL, while replacing the homologous sequence of Bcl-2 by the X domain of Bcl-xL increased Bcl-2 mitochondrial localization [39].

Like we previously did for its proapoptotic partner Bax, we set up cell-free protein synthesis of Bcl-xL, in the absence [35] or in the presence [34] of nanodiscs. After the synthesis, reaction mixes were centrifuged at 20,000× *g* to check for the presence of precipitated protein. Whether in the absence or in the presence of nanodiscs, the vast majority of Bcl-xL was found in the supernatant (Figure 1A). This was different from Bax, which remained soluble in the absence of nanodiscs, but precipitated in their presence [34]. Cell-free synthesized Bcl-xL appeared as several bands. It is now well established that Bcl-xL is the target of nonenzymatic deamidation on residues N52 and N66 located in the unstructured loop between α1 and α2 helices [29,30]. Cell-free protein synthesis conditions are likely favorable for Bcl-xL deamidation [37], because of pH (8.0), temperature (28 °C), and duration (16 h).

Nanodiscs were then purified from the supernatant by affinity chromatography on Ni-NTA. The presence of Bcl-xL was then probed in the flowthrough and bound fractions. We found that Bcl-xL was largely present in the bound fraction, showing that the protein was physically associated with nanodiscs (Figure 1B).

As a negative control, we performed the synthesis of a mutant having a truncation after residue 213, lacking the whole C-terminal α-helix. This deletion was previously shown to impair the mitochondrial localization of Bcl-xL in both mammalian cells [39] and following heterologous expression in yeast cells [26]. We observed that, in the presence of nanodiscs, the truncated mutant Bcl-xLΔC(213) was still recovered in the supernatant; however, contrary to the full-length protein, it was not bound to nanodiscs (Figure 1C). This confirmed that the insertion of Bcl-xL into nanodiscs was dependent on its C-terminal α-helix, which is akin to its insertion into biological membranes [26,39].

### 3.2. Nanodisc-Inserted Bcl-xL Is Full-Length

When produced as a recombinant protein in *E. coli*, Bcl-xL was present under two forms. The full-length protein was found in inclusion bodies, from which it could be resolubilized via a treatment with chaotropic agents (urea or guanidinium chloride) and further purified and reconstituted into nanodiscs using a co-formation method [21]. Before this study, soluble Bcl-xL was usually isolated from the supernatant, and, until 2015, investigators were unaware that this soluble fraction actually displayed a truncation that removed the 13 C-terminal residues. This mistake was caused by the abnormal migration of Bcl-xL on Tris/glycine SDS-PAGE, making both full-length (233 residues) and truncated (218 residues) Bcl-xL display a similar apparent size. As stated by Yao et al., “the finding that soluble Bcl-xL produced in bacteria is cleaved at the C-terminus impacts the many biochemical studies performed with recombinant N-terminal-tagged protein, which was presumed to include all C-terminal residues. Thus, effects attributed to the C-terminal tail may need to be reinterpreted in light of C-terminal truncation” [21]. It should be noted that many biochemical studies have been conducted with recombinant Bcl-xL in which the whole C-terminal α-helix was deleted on purpose to facilitate its production as a soluble protein [19,20,40,41,42,43,44,45,46,47,48]. However, several studies relied on the synthesis of recombinant full-length Bcl-xL purified from *E. coli* lysate-soluble fractions where the C-terminal was likely truncated after residue 218 [46,49,50,51].

We then tested if cell-free synthesis of Bcl-xL would produce a full-length protein or a truncated form. Bcl-xL truncation by bacterial protease(s) after residue 218 removes two C-terminal positively charged residues (R_230_K_231_), which is expected to change the pI of the protein. Both nanodisc-inserted and non-inserted Bcl-xL were analyzed by IEF, and compared to Bcl-xL produced in *E. coli*, recovered either from inclusion bodies (full-length Bcl-xL) or from the soluble fraction (truncated Bcl-xL). We found that nanodisc-inserted Bcl-xL had a similar pI to Bcl-xL present in inclusion bodies (red line), while non-inserted Bxl-xL had a similar pI to Bcl-xL present in *E. coli* soluble fraction (green line) (Figure 2). This suggested that nanodisc-inserted Bcl-xL was actually full-length, while non-inserted Bcl-xL was truncated. We also noticed the presence of an additional form (blue line) having an intermediate pI between full-length and truncated Bcl-xL (after residue 218), which was present in the non-inserted fraction of cell-free synthesis, but not in bacterial lysates. The nature of this form is unknown, and investigations are underway to determine whether its is due to cleavage at a position other than 218, or to a post-translational modification, such as deamidation [29,30]. Interestingly, we observed that co-producing Bcl-xL with its proapoptotic partner Bax (in the absence of nanodiscs) did not protect Bcl-xL against cleavage, since the full-length form was absent, seemingly increasing the proportion of this unidentified intermediate form compared to the truncated form after residue 218 (Figure 2).

To further confirm that nanodisc-inserted Bcl-xL was full-length, a MALDI-MS analysis was performed on the protein. Nanodiscs containing Bcl-xL were bound on His-Trap and then solubilized by adding 50 mM sodium cholate. The eluate contained Bcl-xL and was free of His_7_-MSP1E3D1 (Figure 3A). It was dialyzed to eliminate cholate and concentrated. MALDI-MS analysis showed that the protein had a mass of 26,046 Da (±10 Da), close to the theoretical mass of 26,049 Da for the full-length protein (Figure 3B). As a matter of comparison, the full-length protein purified from *E. coli* inclusion bodies was analyzed in parallel and showed two peaks: a major one with a mass of 25,906 Da (±10 Da) and a minor one with a mass of 26,095 (±10 Da). The mass difference between the mass measured for nanodisc-inserted Bcl-xL and the mass measured for the main peak of the full-length protein purified from *E. coli* corresponds to 140 Da which is consistent with the mass of the initial methionine. The minor peak observed in the protein purified from inclusion bodies might correspond to the full-length protein including the initial methionine modified by an acetylation (+42 Da).

From these data, we could conclude that Bcl-xL synthesized in cell-free assays and inserted into nanodiscs was full-length, without the truncation of the 13 C-terminal residues [21], which would have generated a protein having a lower mass of 24,811 Da. The large width of the peak and the presence of a shoulder at a lower mass suggest that this methionine was also cleaved in a significant proportion of the protein produced in the cell-free system.

### 3.3. Bcl-xL Stimulates the Membrane Insertion of Bax into Nanodiscs

We previously reported that the presence of nanodiscs during the cell-free synthesis of Bax induced its partial precipitation, and that the protein was not inserted into nanodiscs [34]. We then tested if the co-synthesis of Bax and Bcl-xL could help Bax insertion. As we previously reported [34], the co-expression of Bcl-xL prevented Bax precipitation in the presence of nanodiscs (Figure 4A). However, Bcl-xLΔC did not, suggesting that Bcl-xL membrane insertion was needed. We then tested whether Bax and Bcl-xL were actually inserted into nanodiscs. Nanodiscs in the supernatant from the expression of Bax alone or from the co-expression of Bax and Bcl-xL were bound to His-Trap to measure the association of Bax. When expressed alone, Bax was not associated to nanodiscs, whereas, when co-expressed with Bcl-xL, the association of Bax to nanodiscs was significantly increased since about half of the protein was in the bound fraction (Figure 4C,D). As when expressed alone (Figure 1), Bcl-xL was almost completely bound to nanodiscs (Figure 4C).

Nanodiscs from the Bax/Bcl-xL co-expression were then treated with sodium carbonate to measure the insertion of both proteins. Both proteins remained associated to nanodiscs, before or after the treatment, showing that they were actually inserted into nanodiscs (Figure 4E). This stimulation of Bax membrane insertion by Bcl-xL is in line with previous observations showing that the overexpression of Bcl-xL increased Bax localization and insertion into the MOM of yeast cells, mouse fetal liver cells FL5.12, and human colorectal cancer cells HCT-116 [26].

## 4. Discussion

We showed that Bcl-xL could be produced in a cell-free system in the presence of nanodiscs, under conditions where the protein was efficiently inserted (Figure 1). Once inserted, the protein was protected from the cleavage that occurs in *E. coli* (Figure 2 and Figure 3). It may be noted that, in spite of the presence of protease inhibitors, the cleavage likely occurred in the lysate since the migration of non-inserted Bcl-xL on IEF was similar to the migration of the protein recovered from the soluble fraction of lyzed *E. coli* cells, and it was different from the migration of the nanodisc-inserted protein and the protein recovered from *E. coli* inclusion bodies (Figure 2). Interestingly, the co-expression of Bcl-xL with Bax allowed the latter to be inserted into nanodiscs, while Bax was not inserted when expressed alone (Figure 4). This parallels the effect of Bcl-xL overexpression on Bax mitochondrial insertion in yeast and mammalian cells [26].

Since their description in the early 2000s [52], nanodiscs have been widely used to reconstitute membrane proteins, mainly for structural studies. Contrary to detergents, they provide a true membrane environment, and they are far more stable and homogenous in size than liposomes (see [53,54] for reviews). Their size can be adjusted through the use of different MSP variants, whereby the His_7_ tag facilitates nanodisc purification to homogeneity. However, contrary to liposomes, they do not delimitate two compartments and, thus, cannot be used for transport studies.

Nanodiscs have been already used for structural NMR studies on Bcl-xL [55,56]. In the first study [55], Bcl-xL was deprived of the α1–α2 loop (residues 45–84) to facilitate its purification. In the second study [56], the protein was actually full-length (including the C-terminal α-helix) since it was purified from inclusion bodies. However, the purification followed a denaturing protocol, using reverse-phase HPLC. In a more recent study, the same group produced full-length Bcl-xL as a C-terminal fusion with intein that was nondenaturing and kept the C-terminus of Bcl-xL intact [22], a method previously been used by others [57,58]. In all these studies, Bcl-xL was first purified, and then reconstituted in nanodiscs using a co-formation protocol. To our knowledge, our work is the first example of Bcl-xL being spontaneously inserted into nanodiscs, thus limiting conformational alterations that may occur during the purification process.

Bcl-xL’s proapoptotic partner Bax has also been reconstituted in nanodiscs. A tBid-activated monomer (or dimer) was shown to generate a lipidic pore in nanodiscs [59,60]. We recently reconstituted a constitutively active mutant under an oligomeric form [34]. In both studies, Bax was purified before being reconstituted in nanodiscs via co-formation. In the present study, owing to the presence of Bcl-xL, Bax could be directly inserted into preformed nanodiscs, and likely displays a conformation that might be similar to its conformation in cancer cells, where Bcl-xL is overexpressed and both proteins are inserted together into the MOM, as an inactive heterodimer [26]. This setup will be further used to get more precise insight into which residues/domains of the proteins are involved in their interaction in membranes. It is known that, for example, Bcl-xL-G138 and Bax-G67 are involved in their interactions in solution since replacements by larger residues break their interaction [61,62]. Furthermore, the C-terminal residues of Bcl-xL are required for Bcl-xL-mediated Bax retrotranslocation [25], but not for Bcl-xL-mediated Bax/MOM translocation [26]. The combination of cell-free Bax/Bcl-xL co-synthesis in the presence of nanodiscs with site-directed mutagenesis on both proteins will allow a better understanding of how they interact during these processes of translocation, retrotranslocation, and membrane insertion.

## Figures and Tables

**Figure 1 biomolecules-13-00876-f001:**
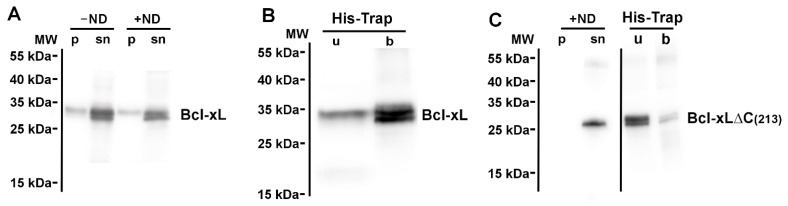
Neosynthesized Bcl-xL is spontaneously inserted into nanodiscs. (**A**) Bcl-xL was synthesized in a cell-free protein synthesis system in the absence (−ND) or in the presence (+ND) of nanodiscs. After a 16 h synthesis, the reaction mix was centrifuged at 20,000× *g* to separate precipitated proteins (in the pellet, p) and soluble proteins and nanodiscs (in the supernatant, sn). Contrary to its proapoptotic partner Bax [34], the presence of nanodiscs did not induce any precipitation of Bcl-xL. The Western blot is representative of more than 10 independent experiments. (**B**) Supernatant from the reaction mix +ND was incubated with Ni-NTA agarose to bind His_7_-tagged MSP1 nanodiscs. After washing, soluble Bcl-xL was found in the unbound fraction (u). After nanodisc elution with imidazole, nanodisc-associated Bcl-xL was found in the bound fraction (b). The Western blot is representative of more than 10 independent experiments. (**C**) The truncated mutant Bcl-xLΔC(213) was produced as in (**A**), and the association to nanodiscs was measured as in (**B**). After cell-free synthesis, the protein was present in the supernatant. However, after Ni-NTA binding, it remained in the unbound fraction, showing that it was not associated to nanodiscs. The Western blot is representative of two independent experiments.

**Figure 2 biomolecules-13-00876-f002:**
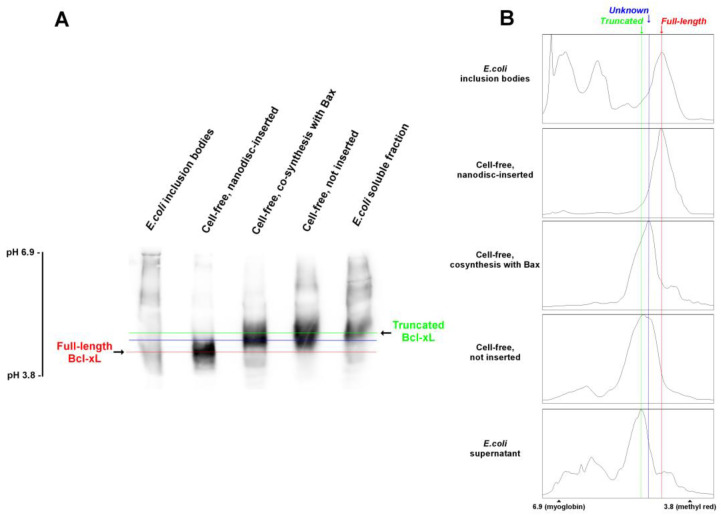
IEF of different forms of recombinant Bcl-xL in different fractions. (**A**) Different fractions of recombinant Bcl-xL were loaded on isoelectrofocusing (IEF) gels (Thermofisher), according to the manufacturer’s instructions. From left to right: Bcl-xL extraction from inclusion bodies in *E. coli* [21,37]; Bcl-xL produced in cell-free system, inserted into nanodiscs (fraction “b” in Figure 1B); Bcl-xL co-produced in cell-free system with its partner Bax, without nanodiscs [34]; Bcl-xL produced in cell-free system, not inserted into nanodiscs (fraction “u” in Figure 1B); Bcl-xL in the soluble lysate from *E. coli* (classical method, before [21]). The blot is representative of two independent experiments. (**B**) Densitometry scans of the lanes shown in (**A**). The red line marks the position of full-length Bcl-xL found in *E. coli* inclusion bodies, which is also found in cell-free synthesized nanodisc-inserted Bcl-xL; the green line marks the position of truncated (after M218) found in *E. coli* soluble fraction (Yao et al., 2016) and in non-inserted cell-free synthesized Bcl-xL. The blue mark corresponds to an unknown form, which might have been generated by truncation at another position or by a post-translational modification such as deamidation. This form was only found in non-inserted cell-free synthesized Bcl-xL, but not in *E. coli* or in nanodisc-inserted cell-free synthesized Bcl-xL.

**Figure 3 biomolecules-13-00876-f003:**
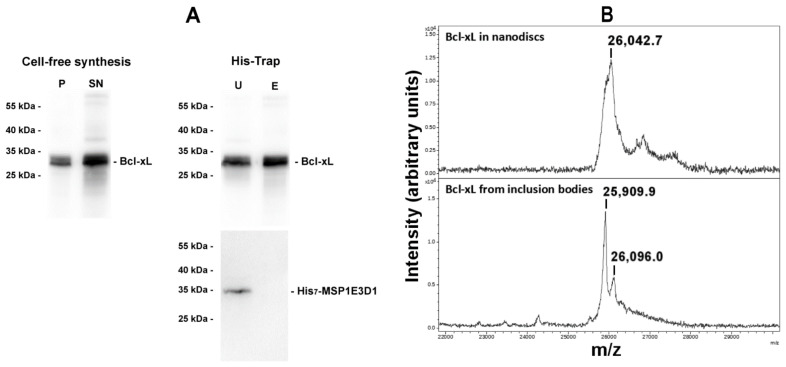
MALDI-MS analysis of Bcl-xL shows that the nanodisc-inserted protein is full-length. (**A**) Purification of nanodisc-inserted Bcl-xL. Bcl-xL was produced in the cell-free system in the presence of nanodiscs as in Figure 1A (**left** panel) and then loaded on His-Trap resin (**right** panel). The unbound fraction was washed (U), and the resin was added with 50 mM cholate to solubilize nanodiscs (E). The “E” fraction contained Bcl-xL but not His_7_-MSP1E3D1. (**B**) MALDI-MS analysis of the “E” fraction from above (**top**), compared to full-length Bcl-xL purified from inclusion bodies of *E. coli* [21,37] (**bottom**). Nanodisc-inserted Bcl-xL displayed a major peak at 26,046 Da (±10 Da) (mean of two individual measurements), close to the theoretical mass of full-length Bcl-xL (26,049 Da), while Bcl-xL purified from inclusion bodies displayed two peaks at 25,906 Da and 26,095 Da (mean of two individual measurements), close to the theoretical mass of full-length Bcl-xL deprived of the initiation methionine (theoretical mass of 25,917 Da) and to the full-length protein carrying an acetylated methionine (theoretical mass of 26,091 Da), respectively. Note that minor peaks with higher masses in nanodisc-inserted Bcl-xL, might correspond to termination errors of the UGA (opal) codon that might happen in cell-free synthesis.

**Figure 4 biomolecules-13-00876-f004:**
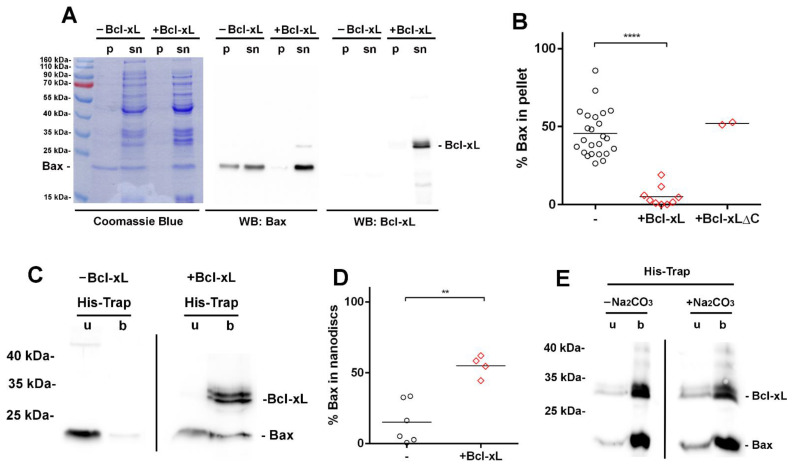
Effect of Bcl-xL on Bax membrane insertion. (**A**) Bax was expressed alone (−Bcl-xL) or co-expressed with Bcl-xL (+Bcl-xL) in the presence of nanodiscs. After the synthesis, the reaction mix was centrifuged at 20,000× *g* (10 min) to recover a pellet (p) containing precipitated proteins and a supernatant (sn) containing soluble proteins and nanodiscs. We observed that Bcl-xL prevented the partial precipitation of Bax. (**B**) Quantification of the effect of Bcl-xL and Bcl-xLΔC_(213)_ on Bax precipitation. Non-saturated Western blots obtained in (A) were quantified with Image J. Column statistics: −Bcl-xL: 45.5 ± 14.9 (SD) (*n* = 24), +Bcl-xL: 5.1 ± 6.3 (SD) (*n* = 9), +Bcl-xLΔC: 52.0 ± 1.2 (*n* = 2); **** *p* < 0.0001. (**C**) Nanodiscs from the expression of Bax alone (−Bcl-xL) or from the co-expression of Bax and Bcl-xL (+Bcl-xL) were bound to His-Trap. The bound fraction (b) contained proteins associated to nanodiscs, while the unbound fraction (u) contained soluble proteins. The co-expression with Bcl-xL induced the partial association of Bax to nanodiscs. (**D**) Quantification of the effect of Bcl-xL on Bax interaction with nanodiscs. Non-saturated Western blots obtained in (**C**) were quantified with Image J. Column statistics: −Bcl-xL: 15.1 ± 14.9 (*n* = 6), +Bcl-xL: 54.9 ± 7.6 (*n* = 4); ** *p* < 0.01. (**E**) Nanodiscs from the Bax/Bcl-xL co-expression were purified on His-Trap (**B**), treated or not with 0.1 M sodium carbonate (pH 10.0; 10 min), and repurified on His-Trap. The unbound fraction (u) contained proteins loosely interacting with nanodiscs, while the bound fraction (b) contained membrane-inserted proteins. The vast majority of Bcl-xL and Bax proteins were membrane-inserted.

## Data Availability

The data presented in this study are available on request from the corresponding authors.

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
