# Peer review of "Bcl-xL Is Spontaneously Inserted into Preassembled Nanodiscs and Stimulates Bax Insertion in a Cell-Free Protein Synthesis System"

_biomolecules, 2023, doi:10.3390/biom13060876_

Round 1
Reviewer 1 Report
Eyitayo, AR., et al have evaluated a model system combining cell-free protein synthesis and nanodisc insertion to study the molecular aspects of the interaction between the anti-apoptotic protein Bcl-xL (especially the C-terminal α-helix), a major regulator of cell death and survival and Bax, a pro-apoptotic partner, during their membrane (nanodisc) insertion to gain further information about its membrane organization of the mitochondrial outer membrane (MOM).
The authors demonstrated that opposite to Bax, neo-synthesized Bcl-xL was instinctively inserted into nanodiscs, whereas deletion of the C-terminal α-helix of Bcl-xL prevented nanodisc-insertion. Moreover, Bcl-xL got shielded against the proteolysis of the 13 C-terminal residues due to nanodisc-insertion. In this study, the authors concluded that cell-free synthesis in the presence of nanodiscs is thus a suitable model system to contemplate the molecular basis of the Bcl-xL – Bax interaction during membrane insertion.
The authors provide evidence that the full-length Bcl-xL can be simultaneously synthesized and inserted into nanodiscs, and further stimulates Bax insertion. Moreover, the data indicate that Bcl-xL could be produced in a cell-free system, inserted, and protected in the nanodiscs. The co-expression of Bcl-xL with Bax allowed the latter to be inserted into nanodiscs, mimicking the effect of Bcl-xL overexpression on Bax mitochondrial insertion. These results provide some evidence that cell-free synthesis in the presence of nanodiscs is an adequate model to study the molecular mechanisms driving the membrane insertion of Bcl-xL and its effects on Bax insertion.
The experiments designed for this study are justified. Overall, the results are significant, and the introduction was written very carefully with proper references and explained significance which is highly appreciated. There is a lack of writing and/or presentation arrangements. The authors need to explain a few important topics like nanodisc, Bax function, etc. rather than providing only references. Nonetheless, the article seemed to possess significant value towards molecular aspects of BcL-xL and Bax interaction. The novelty of this study as demonstrated by the authors is that Cell-free synthesis in the presence of nanodiscs is a convenient model system for the study of BcL-xL and Bax membrane insertion.
Overall, the clarity of the text is good and needs a few fine readjustments. The manuscript has a few typographical and grammatical errors. The main figures were consistent with proper legends. In general, the manuscript can accomplish the caliber of quality for consideration for publication in the Journal of Biomolecules with few details. The authors are advised to consider the comments below:
Major comments
1. One of the major flaws is writing the methods without subdivision. It is very difficult to understand or read the methods and highly recommendable to re-write the method portion in parts like:
2. Materials and Methods
2.1 Cloning of Full-length Bcl-xL and Bax
2.2 Production of the scaffold protein His7-MSP1E3D1
2.3 Nanodisc preparation……………
2. The abbreviation of “ND” (Figure 1) is not mentioned all over the text.
3. Please provide details of affinity chromatography on Ni-NTA.
4. Please provide appropriate references for this statement/page 7 / lines 230-232 / “It follows that, until 2015, many in vitro studies with purified Bcl-xL were actually done with the truncated protein” (or with a mixture of both forms).
5. The abbreviation of “IEF” (Figure 2) is not mentioned all over the text.
6. Molecular weight markings are missing in Figure 4A gels.
7. “Nanodisc” is a major topic in this manuscript that needs to be discussed as to how it is relevant to the expression of Bcl-xl and Bax.
8. I would request to separate the result and discussion parts separately.
Minor comments
1. Page 2/ line 35 / It seems that interaction may not agree in number with other words in this phrase.
2. Page 2 / Line 45/ Check spelling “conformational”
3. Page 2 / Line 62/ Check spelling “questionning”
4. Page 5 / Line 175/ Check spelling “aknowledged”
5. Page 7 / Line 219/ Check spelling “Howewer”
6. Figure 4C / Check the subscript of “Na2CO3”
Author Response
please see the attachement

Reviewer 2 Report
In this manuscript, the authors propose a cell-free system that utilizes nanodiscs to study the molecular interactions between BCLXL and BAX during membrane insertion. Specifically, they report that Bcl-xL can be spontaneously inserted into preassembled nanodiscs and further stimulates Bax insertion in the system. The topic of this study is of significant interest in the field, and the manuscript is well written, with a descriptive introduction and methods that outline the state of the art and rationale of the work. However, there are some aspects of the manuscript that require clarification and addressing of concerns.
Major points:
1- One of the main results of this study is the simultaneous synthesis and insertion of full-length Bcl-xL into nanodiscs, which stimulates Bax insertion. However, a similar result was previously reported by the same authors, wherein Bcl-xL increased the insertion of BaxWT into nanodiscs (reference 34, Figure 1F; https://doi.org/10.1016/j.bbamem.2022.184075). The authors should clarify the differences between both studies and highlight the novel contributions of this current study.
2- The authors should extend the discussion to include the advantages of the system, comparisons with other systems, and possibilities for studying other members of the BCL-2 family.
3- In Figure 2, the bands in the IEF gel related to E. coli inclusion bodies are not clearly evident.
4- The authors only specify that the results in Figure 3B are means of two individual measurements. It is unclear if the other results were only performed once. The authors should specify the “n” of the experiments.
5- The authors state that the association of Bax to nanodiscs was significantly increased by approximately half of the protein, but they do not quantify or perform statistics. The authors should quantify and perform statistics on some of the experiments, especially those with quantitative results, as they have done in previous publications.
Minor points:
In line 325, the authors describe point 4 as a discussion, but it appears to be a conclusion.
Round 2
Reviewer 2 Report
The authors have satisfactorily answered all my questions and have taken my recommendations into account when improving the figures and editing the current version of the manuscript. In my opinion the manuscript can be published in its current form.